# Molecular insights into heart field-specific cardiomyocyte differentiation - A computational study

Ricco Zeegelaar❖, Georgios Argyris❖, Janine N. Post* 

Quantitative Biology Lab, Developmental BioEngineering, Faculty of Science & Technology, University of Twente, Enschede, The Netherlands

❖ These authors contributed equally to this work.
* j.n.post@utwente.nl

## Abstract

Understanding the mechanisms underlying cardiomyocyte (CM) differentiation is essential for the accurate generation of the different types of heart cells in vitro. This study advances current models of CM differentiation by introducing a gene regulatory network (GRN) model that integrates early heart field formation with downstream differentiation of committed cardiomyocytes into atrial and ventricular subtypes. The model is implemented using Boolean logic, enabling qualitative simulation of cardiac regulatory dynamics. Attractor analysis identifies steady states corresponding to first and second heart field derived atrial and ventricular cardiomyocytes. The model reveals the mechanism of WNT and BMP signaling in heart field determination and shows how RA regulation of *NR2F2* decisively determines atrial versus ventricular cardiomyocyte cell fate. The model reproduced published knockout and overexpression experiments, and probabilistic simulations estimate differentiation efficiencies under varying signaling inputs. The unified Boolean model provides a foundation for generating heart-field-specific cardiomyocytes with precise atrial or ventricular identities, supporting efforts in directed differentiation and targeted heart cell therapies.

## Introduction

The heart is composed of several different cell types, including endothelial cells, fibroblasts, and cardiomyocytes (CMs), which all arise from mesodermal progenitors during embryogenesis [1]. Early cardiac development initiates with the formation of two distinct heart fields: the first heart field (FHF) and the second heart field (SHF). These heart fields guide early heart morphogenesis and structural compartmentalization of the heart [2,3]. Both heart fields contribute to the formation of cardiomyocytes that populate the atrial and ventricular chambers [4,5] (Fig 1). Atrial and ventricular cardiomyocytes (aCMs and vCMs, respectively) are not only functionally distinct but

**Data availability statement:** All relevant code and simulations are available on Github: github.com/GeorgiosArg/Beyond-Committed-cardiomyocytes.

**Funding:** This work was supported by the Dutch Research Council (NWO), NWO-XL grant number OCENW.XL21.XL21.067 to JNP (https://www.nwo.nl/). The funder had no role in study design, data collection and analysis, decision to publish, or preparation of the manuscript.

**Competing interests:** The authors have declared that no competing interests exist.

also exhibit differences in gene expression profiles, including differential activation of key transcriptional regulators [5,6].

Many congenital and acquired heart diseases affect specific regions of the heart [1]. For instance, myocardial infarction commonly results in irreversible loss of cardiomyocytes primarily localized to the left ventricle (LV), a region predominantly populated by cells originating from the FHF [9]. For cell replacement therapies to be effective, transplanted cardiomyocytes must integrate with host tissue. FHF-derived ventricular cardiomyocytes are optimal candidates for treating LV cell loss, as their shared developmental origins enhance electrical coupling and reduce the risk of post-remodeling arrhythmias [10,11]. This lineage-dependent therapeutic potential highlights the need to distinguish not only between atrial and ventricular subtypes, but also between cardiomyocytes of distinct heart field origin.

Gene regulatory networks (GRNs) control cell fate decisions during differentiation by coordinating the expression of key genes in response to internal and external signals. In the context of cardiomyocyte differentiation, numerous regulators have been identified, including transcription factors NKX2.5, TBX5, HAND1, HAND2, ISL1, and TBX1, as well as signaling pathways FGF, BMP, retinoic acid (RA) and WNT [12,13]. The critical roles that these factors individually play in establishing atrial and ventricular identities are well established. There however is limited understanding of the emergent lineage decision mechanisms that these factors within GRNs give rise to, particularly governing heart field specification and cardiomyocyte subtype differentiation. Especially how early lineage decisions influence the final phenotype of the cell, remains incompletely characterized. This gap in understanding limits the ability to generate CMs with defined heart field and chamber-specific identities in vitro, thereby constraining the development of targeted regenerative therapies.

Computational modeling offers a means to investigate the emergent behavior of GRNs. A variety of modeling formalisms is available for modeling biological systems, such as Boolean networks, Bayesian networks, agent-based models and rule-based models to name a few (as reviewed in [14]). The modeling strategy to use is dependent on the system to model and the type of data available and can be classified between mechanistic and empirical approaches. Mechanistic models can simulate the dynamic behavior of regulatory systems but typically depend on detailed kinetic parameters and large-scale quantitative datasets, which are often unavailable in developmental systems. Empirical models infer network structure from experimental data but mainly capture associations between network components without their direct regulatory relationships [15,16]. Boolean network (BN) models offer a practical alternative by representing gene regulatory interactions with binary states and logical rules [15,16]. We can use these Boolean networks to make predictions of cardiac differentiation trajectories and to predict the molecular conditions required for generating heart-field-specific cardiomyocytes, without requiring extensive quantitative data.

A previous Boolean network model by Hermann et al. [4] simulated early cardiac development by modeling regulatory interactions governing FHF and SHF specification. This model identified two steady states corresponding to distinct heart field identities, but did not incorporate the downstream differentiation of committed

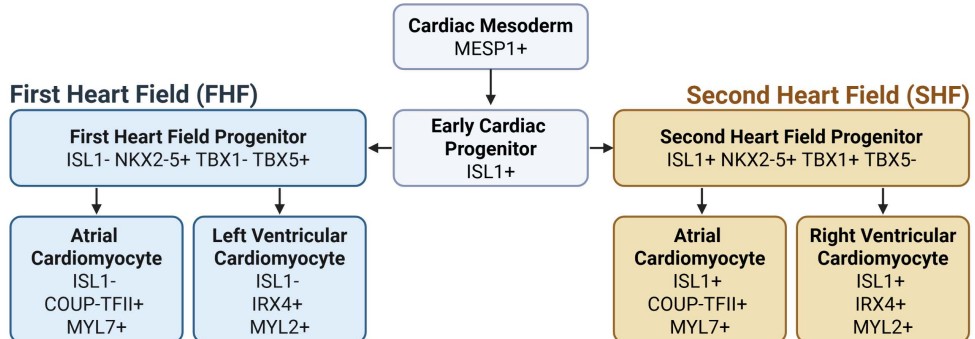

**Fig 1. Developmental origins of cardiomyocyte subtypes from cardiac mesoderm.** Early cardiac mesoderm is induced by transcription factor MESP1 Early cardiac progenitors initially express ISL1 and give rise to both the first heart field (FHF) and the second heart field (SHF). These lineages are distinguished by differential marker gene expression: FHF progenitors downregulate ISL1 and express TBX5, while SHF progenitors maintain ISL1 expression and express TBX1. The FHF contributes primarily to left ventricular CMs and a subset of atrial cardiomyocytes, while the SHF gives rise to right ventricular CMs, the outflow tract and additional atrial CMs MESP1 [7,8]. In this article, atrial and ventricular cardiomyocyte identities are defined based on model outputs: atrial cardiomyocytes by expression of COUP-TFII and MYL7, and ventricular cardiomyocytes by expression of IRX4 and MYL2.

cardiomyocytes into atrial and ventricular subtypes. Extending this framework to include later stages of differentiation is essential for capturing the full progression of cardiomyocyte development and for enabling predictive simulations of cardiomyocyte subtype generation in the context of targeted regenerative therapy.

This study presents an extended Boolean network modeling framework that integrates early heart field specification with the subsequent differentiation of committed cardiomyocytes into atrial and ventricular subtypes. By combining the heart field identity model from Herrmann et al. [4] with a novel cardiomyocyte subtype differentiation network, the model captures how key signaling pathways govern sequential stages of development. The unified network identifies four steady states corresponding to heart-field-specific cardiomyocyte subtypes. Simulations demonstrate the model's ability to predict differentiation outcomes and efficiency based on combinations of signaling inputs. This framework provides a mechanistic basis for the coupling between early and late-stage molecular regulation that determines cardiomyocyte subtype identity and enables prediction of signaling conditions required to generate heart field-specific cardiomyocyte subtypes in vitro.

## Materials and methods

To investigate the regulatory mechanisms of heart-field-specific cardiomyocyte differentiation, a new Boolean network (BN) model was constructed to describe the differentiation of committed cardiomyocytes into atrial and ventricular subtypes. This model was integrated with a previously published BN representing heart field specification [4], resulting in a unified gene regulatory network (GRN) capable of simulating heart-field-specific cardiomyocyte subtype formation. The following sections describe the construction of the new model, the integration strategy, and simulation approaches used to analyze and validate network dynamics.

### Network construction

The cardiomyocyte subtype Boolean network was constructed through literature-based identification of key regulatory factors involved in atrial and ventricular cardiomyocyte differentiation. Transcription factors and signaling molecules identified from mouse and human studies were represented as nodes, and experimentally supported regulatory interactions were implemented as directed edges, categorized as either activating or inhibitory. Only interactions with direct experimental

support in cardiomyocytes – derived from either mouse or human studies – were included in the model. The network was gradually expanded, and interactions were refined until simulations reproduced known gene expression dynamics during cardiomyocyte specification, and then reduced to a minimal structure that preserved this behavior. A complete list of included interactions and their literature sources is provided in S1 Table.

The cardiomyocyte subtype network was integrated with the Boolean model of heart field identity from Herrmann et al. [4], which exhibits two steady states corresponding to the first heart field (TBX5+) and the second heart field (ISL1 + TBX1+). The integration was achieved by linking output nodes of the heart field model – specifically GATA4/6 and NKX2.5 – to corresponding input nodes in the cardiomyocyte subtype differentiation model. These connections were incorporated as direct inputs in the update functions of the downstream nodes, including IRX4 and NR2F2, reflecting experimentally supported regulatory dependencies. The logical rules were adjusted to include the upstream inputs, while preserving the behavior of the original network. The update functions of the cardiomyocyte subtype model, the adapted heart field network, and the unified model network with added interactions are provided in S2A–S2C Table.

## Model dynamics

The nodes of the model are equipped with Boolean dynamics, each taking a value of 0 or 1 to indicate whether the node is active or inactive. The state of the network at a given time point is represented by a vector containing the activity of all nodes. Node states are updated according to logical update functions, which determine the value of each node based on the current values of its regulators. For each consecutive time step, the network will therefore transition into a new state.

Repeated application of the update functions generates a sequence of states and transitions, referred to as a simulation (in literature, such sequences are also termed solutions, orbits, and trajectories, particularly in continuous-time dynamical systems). These simulations form the basis to analyzing network dynamics.

All simulations were encoded into a State Transition Graph (STG) which contains all possible states and all transitions between the states. For a BN with n nodes, the STG contains $2^n$ states. The updates of the STG can be performed synchronous – where all variables are updated at the same time – or asynchronous, in which only one variable is non-deterministically (randomly) selected and updated as exemplified in Fig 2. In this study, the asynchronous update mode was used, as it is one of the most biologically plausible update modes due to the different timescales of biological processes [17].

When no specific time delays are defined, as in this study, asynchronous updating generates all possible transitions between states. Over time, the network will transition into a state where there is no successor state distinct from itself. These are referred to as steady or attractor states. In BNs involving cell differentiation processes, the steady states generally correspond to distinct cell types.

$$x(t+1) = y(t) \lor z(t)$$
$$y(t+1) = x(t) \land z(t)$$
$$z(t+1) = \neg x(t)$$

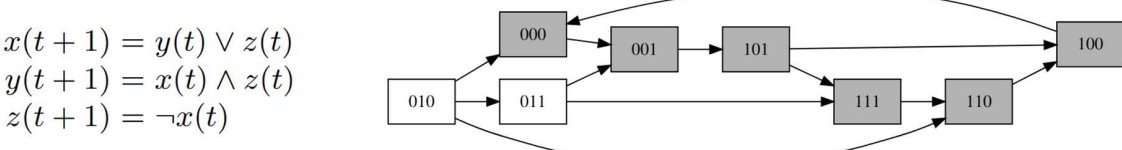

**Fig 2. (Left) Schematic example of a Boolean network (BN) with update functions of three nodes (x,y and z). (Right) The corresponding state transition graph (STG) generated by the asynchronous update mode wherein only one variable is non-deterministically selected and updated at each time step. Self-transitions are not displayed for clarity.**

## Model validation

Boolean networks of gene regulatory networks can be validated internally and externally. When a GRN is generated using data, internal validation is performed by determining the quality of the data, and the performance of the inference algorithm [18]. The networks presented in this paper are constructed from published node interactions (see S1 Table and Herrmann et al. [4]), causing included interactions to already be internally validated. The model can also be validated externally by comparing the model topology to a data-generated BN model, or by comparing model predictions with biological data or observations [18]. A data-generated BN can for example be generated using the R package BoolNet [19]. However, this requires a dataset with inputs and readouts that match the model, which does not exist for the network presented in this paper. Additionally, a data-generated BN is not expected to exhibit biologically plausible steady states that capture interactions across two separate time steps, which is necessary for comparison to the presented BN. The best method for externally validating the presented BN is therefore to compare model predictions to external information not used during the modeling process [18]. The model will be qualitatively validated to see if it can predict expression outcomes of published knockout and overexpression experiments, with the methodology exemplified in the section "Knockouts and overexpression for validation".

Model predictions are made by computing the attractors of the asynchronous BN using the tool Boolsim [20], incorporated in the CoLoMoTo Notebook [21]. All code and simulations are available on:

github.com/GeorgiosArg/Beyond-Committed-cardiomyocytes.

**Knockouts and overexpression for validation.** Because no dataset directly matches the model's inputs with information on node activity, the model topology was qualitatively validated. Literature on knockout and overexpression experiments that influence cardiac development was gathered. These perturbations were then replicated in the model to see if the model predictions still match experimental outcomes.

Gene knockouts were simulated by perturbing the corresponding node value to zero (off-state), while constitutive expression of a gene was modeled by fixing the node to one (on-state). For each perturbation, the model was simulated under the same signaling conditions as in the wild-type simulations (WNT, RA; ON or OFF). It is then identified if the model can still reach the attractor states (which corresponds to a cell type) with the perturbation. These results are compared to the published experimental outcomes. A perturbation is considered to be successfully reproduced by the model if the presence or absence of an attractor state matches the (wet lab) experimentally observed presence or absence of a cardiomyocyte cell type.

**Probabilistic simulations for differentiation efficiency.** To investigate how many cells in a heterogeneous cell population will reach a specific cell state, a probabilistic version of the Boolean network was employed. In this implementation, each transition from a predecessor state to the immediate successor state occurs through the random selection of a single node for updating. For instance, in a network with three variables (as illustrated in Fig 2), each variable has one-third probability to be updated at each time step, resulting in a Markov chain as shown in Fig 3.

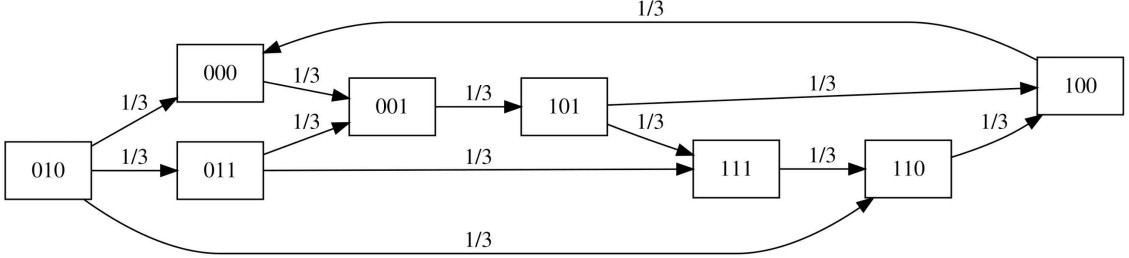

**Fig 3. Markov chain of the BN shown in Fig 2.** At each time step, one variable is selected at random for updating. Each state has exactly 3 outgoing transitions (for clarity, the self-transitions of the states are not displayed). In contrast to the non-deterministic STG of Fig 2, where the BN can occupy several states at a particular time point, here the BN is in exactly one state but with a specified probability.

Repeated simulations from randomized initial conditions will allow estimation of the probability of cells reaching a specific steady state.

The same probabilistic simulation approach was applied for the knockout and overexpression perturbations. The resulting steady-state distributions were compared to reported experimental observations of the corresponding genetic perturbations. By comparing the simulated proportion of cells reaching the steady state associated with specific heart-field-specific cardiomyocyte subtypes to experimentally observed cell-type yields under the same conditions, the model's predictive capacity was further evaluated.

**Transformation from Boolean to real dynamics for robustness testing.**  To assess the robustness of differentiation outcomes under continuous input variations, the BN model was extended to a Boolean Network Extension (BNE), following the methodology of Grieb et al. [22]. In this framework, node activity is represented by continuous values in the interval [0, 1], replacing the original binary (0 or 1) states. Boolean logic update functions from the original BN were transformed into real-valued update functions using the transformation rules shown in Equation 1.

$$x \wedge y \equiv x \cdot y$$

$$x \vee y \equiv x + y - x \cdot y$$

$$\neg x \equiv (1 - x) \tag{1}$$

Equation 1: The transformation of the Boolean rules into real-valued functions. The equivalences provide identical results to their Boolean counterparts when variables are binary. When variables are initialized in the continuous interval [0, 1], the resulting real valued functions exhibit additional behaviors not captured by the original Boolean rules. Application of these transformations yields a discrete-time dynamical system known as a Boolean Network Extension (BNE), as described by [22].

The extension was used to assess the stability of the atrial and ventricular cardiomyocyte states under varying levels of Retinoic Acid (RA) signaling. The RA pathway was selected due to its decisive role in directing atrial versus ventricular specification during cardiac development [13,23]. To assess robustness, RA input levels were systematically varied across the full range [0, 1], while maintaining other signals relevant for early heart development at constant activation levels, consistent with biological conditions. This analysis enabled the characterization of signaling thresholds required for stable differentiation into atrial or ventricular cardiomyocyte states and provided additional validation of the network's predictive capability under continuous input variations.

## Results

### A gene regulatory network of cardiomyocyte specification

The cardiomyocyte subtype Boolean model captures the regulatory pathways and transcription factors involved in the differentiation of committed cardiac precursors into ventricular and atrial cardiomyocytes (Fig 4). The key nodes in the network play crucial roles in guiding CM differentiation. GATA4/6 and the NOTCH signaling pathway are modeled as constitutively active in precursor cells, as they are essential for early-stage development of both atrial and ventricular CMs [4,24–26]. COUP-TFII (encoded by *NR2F2*) is regulated by retinoic acid (RA) and has been identified as a molecular switch that determines whether the committed cell becomes an atrial or ventricular CM. Multiple biomarkers for these cell fates have been identified, which are mostly coregulated and co-expressed [6]. We therefore model these clusters of biomarkers with expression of two representative genes; MYL7 for atrial identities and MYL2 for ventricular identities. When RA is present, *NR2F2* expression is activated and represses ventricular markers like MYL2 while activating expression of atrial markers such as MYL7, which steers the cell toward atrial differentiation. Conversely, when RA is absent, *NR2F2* is not expressed, which allows the differentiation towards ventricular CMs [13,23].

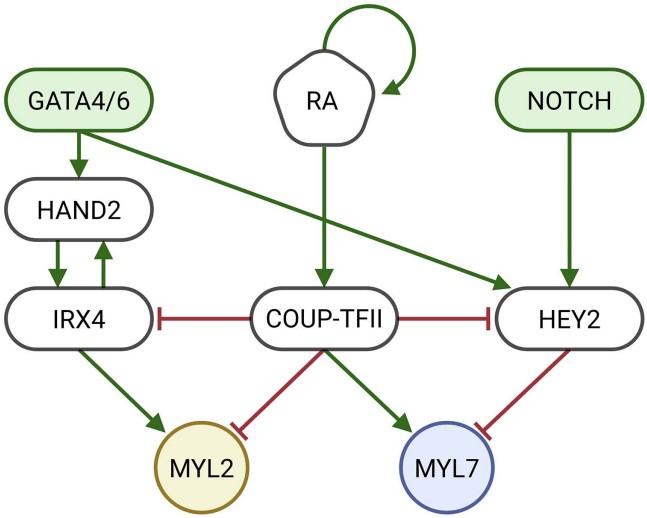

**Fig 4. Key gene regulatory network underlying cardiomyocyte subtype specification.** The schematic shows key signaling interactions involved in atrial and ventricular cardiomyocyte differentiation. Nodes represent signaling inputs, transcription factors and cell identity markers, with colors for easier interpretation. Edges indicate experimentally supported regulatory interactions, with green arrows showing activation and red T-bars marking inhibition. MYL7 (blue) is an atrial biomarker and MYL2 (yellow) is a ventricular biomarker. GATA4/6 and NOTCH (green) are modeled as constitutively active to reflect early developmental conditions. Sources for all interactions are provided in S1 Table, and the Boolean update functions are listed in S2B Table. The steady states of the cardiomyocyte subtype GRN are provided in Table 1.

The steady states obtained from attractor analysis of the cardiomyocyte specification model are presented in Table 1. The states align with established marker gene profiles for ventricular and atrial CMs [13,27]. The BN exhibits two steady states: one steady state with activity of ventricular cell fate indicators IRX4, HEY2, and MYL2, and a second steady state where atrial cell fate indicator COUP-TFII and MYL7 are active. The outcome of the simulation depends on the initial conditions of the network, particularly the activity of retinoic acid (RA), which drives the system toward either atrial or ventricular specification.

## Cardiomyocyte subtype differentiation under varying RA signaling

The Boolean formalism only allows for binary states. In cells however, signaling strength can vary and is dose-dependent. To investigate if the cardiomyocyte specification model can predict what happens under varying signaling strengths,

**Table 1. Simulations of the cardiomyocyte subtype BN result in two steady states.**

|  | Signaling (molecule) | | Transcription factor | | | | | Cell fate indicator | |
|---|---|---|---|---|---|---|---|---|---|
|  | NOTCH | RA | GATA4/6 | COUP-TFII | HAND2 | IRX4 | HEY2 | MYL2 | MYL7 |
| aCM type | 1 | 1 | 1 | 1 | 1 | 0 | 0 | 0 | 1 |
| vCM type | 1 | 0 | 1 | 0 | 1 | 1 | 1 | 1 | 0 |

The rows display the two steady states identified through asynchronous simulation of the Boolean network. Each row corresponds to a steady state, with the columns indicating the activity (1 = active, 0 = inactive) of each network node. During simulations, GATA4/6 and NOTCH were fixed as active (1) to reflect their activity in cardiac progenitor cells. When RA is initially active (RA = 1), the model evolves and stabilizes at a state where MYL7 is active (1) and MYL2 is inactive (0), consistent with an atrial cardiomyocyte identity as observed in vitro [13,27]. In contrast, if the simulation starts from a state where RA is inactive (RA = 0), the model stabilizes in a ventricular state where MYL2 is active (1) and MYL7 is inactive (0), consistent with a ventricular cardiomyocyte identity as observed in vitro [13,27].

the Boolean network was extended into a Boolean network Extension (BNE), following the methodology of [22]. In this framework, binary states (0 or 1) are converted to continuous values in the interval [0, 1], and logical update functions are transformed into real-valued expressions. The signaling inputs GATA4/6 ($\gamma$), RA ($\rho$), and NOTCH ($\nu$) were modeled as continuous parameters to assess their effect on downstream gene activity and differentiation outcomes. The activity of nodes is then expressed as functions of $\rho$, to see how varying levels of RA signaling influences cardiomyocyte identity.

Following transformation to the continuous model, some nodes stabilized to expressions dependent solely on the input parameters. The node of NR2F2 was directly stabilized according to RA with the update function $x_{NR2F2}(t+1) = \rho$. The update function of xHEY2 takes the form $x_{HEY2}(t+1) = \nu \cdot \gamma \cdot (1-\rho)$, indicating dependency on NOTCH, GATA4/6, and inversely RA signaling. The atrial marker MYL7 stabilized to the function $x_{MYL7}(t+1) = \rho \cdot (1 - (\nu \cdot \gamma \cdot (1-\rho)))$, which showcases the switch-like behavior of atrial specification dependent on RA signaling intensity.

In contrast, the nodes HAND2, IRX4 and MYL2 form a coupled linear system of update functions (Equations 2). These nodes do not stabilize independently but rather depend on each other's steady state values.

$$x_{HAND2}(t+1) = x_{IRX4}(t) + \gamma - \gamma \cdot x_{IRX4}(t)$$

$$x_{IRX4}(t+1) = x_{HAND2}(t) \cdot (1-\rho)$$

$$x_{MYL2}(t+1) = x_{IRX4}(t) \cdot (1-\rho) \tag{2}$$

Solving this system at steady state (where $Xi\,(t+1) = Xi\,(t)$), yields expressions at steady states for each variable as a function of the input parameters (Equations 3). These solutions provide insight into how RA, GATA4/6 and NOTCH signaling regulate the balance between atrial and ventricular specification via the identified gene regulatory network.

$$x_{NR2F2} = \rho$$

$$x_{HAND2} = \frac{\gamma}{\rho + \gamma - \rho \cdot \gamma}$$

$$x_{IRX4} = \frac{\gamma}{\rho + \gamma - \rho \cdot \gamma} - \rho \cdot \frac{\gamma}{\rho + \gamma - \rho \cdot \gamma}$$

$$x_{MYL2} = \left( \frac{\gamma}{\rho + \gamma - \rho \cdot \gamma} - \rho \cdot \frac{\gamma}{\rho + \gamma - \rho \cdot \gamma} \right) \cdot (1-\rho)$$

$$x_{HEY2} = \nu \cdot \gamma \cdot (1-\rho)$$

$$x_{MYL7} = \rho \cdot (1 - (\nu \cdot \gamma \cdot (1-\rho))) \tag{3}$$

Below a RA signaling threshold, MYL2 becomes more active, indicating ventricular specification. The relationship between RA signaling intensity and marker activity is shown in Fig 5. These results highlight the robustness of cardiomyocyte subtype predictions and support the role of RA signaling as a switch-like regulator in controlling the differentiation towards atrial and ventricular cardiomyocyte identities.

## Unified Boolean network reveals the path to heart field-specific cardiomyocytes

To construct the unified Boolean network, the previously published model by Herrmann et al. [4], which describes gene interactions during early murine cardiac development, was adapted. This model distinguishes between the FHF and SHF based on specific gene expression profiles. The adapted model, shown in Fig 6, includes key signaling pathways

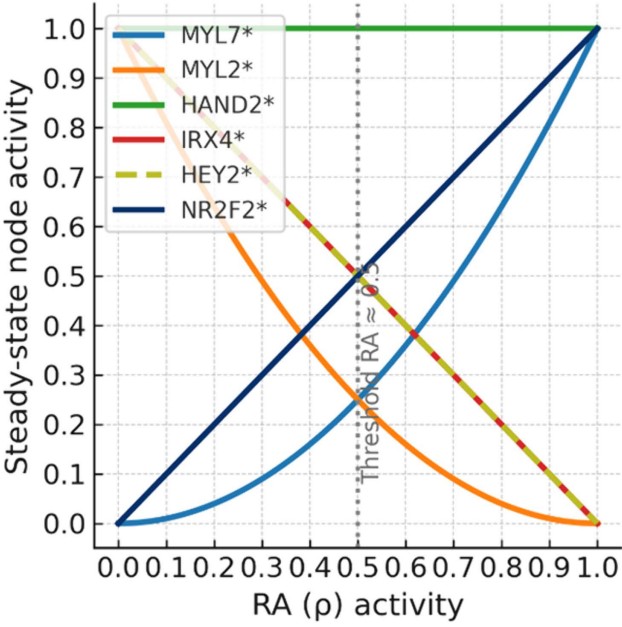

**Fig 5. Steady state activity levels of key network nodes as a function of retinoic acid (RA) signaling intensity.** The X-axis represents the input value of RA (ρ). The Y-axis indicates the steady state activity of nodes, derived from the Boolean Network Extension (BNE). The plot shows that RA orchestrates steady state node activity and therefore cardiomyocyte subtype identity in a signaling intensity-dependent manner. This is consistent with in vivo and in vitro observations, where cardiac subtype fates are dependent on RA concentration [23].

implicated in early heart field identity formation such as BMP, FGF and WNT, as well as key transcription factors determining heart field identity like ILS1, TBX5, and NKX2.5. In the original model, multiple exogenous nodes were used to account for the timed activation of BMP2 (exBMP2) and WNT (exWNT) signaling by neighboring tissues. In the adapted version, the structure is adjusted by using a single exogen to represent the effects of these signals without considering timing or origin.

The unification of the two BNs was achieved by merging the GATA nodes from both the heart field and the cardiomyocyte subtype model and introducing an additional regulatory link between NKX2.5 and IRX4, as supported by experimental evidence [6]. The unified network, shown in Fig 7, incorporates both networks with the additional regulatory interactions, enabling the simulation of cardiomyocyte subtype differentiation downstream of heart field specification. The corresponding update functions are provided in S2C Table.

The steady states of the unified model are derived through attractor analysis, as shown in Table 3. Importantly, since the Boolean network integrates regulatory interactions from multiple stages of differentiation, node activity in steady states should be interpreted in the context of the differentiation stage in which the node is involved, rather than as a direct indicator of transcriptional activity in the final stage of differentiation. For instance, the activity of nodes associated with heart field specification reflects their activity at the heart field specification stage of differentiation and does not necessarily correspond to their activity in the final cardiomyocyte subtype. Based on this interpretation, the steady states reveal which pathways and transcription factors were active at key regulatory checkpoints during differentiation, influencing the final cell fate. To support interpretation, nodes have been annotated with their associated cell type, clarifying their stage specific roles in cardiomyocyte subtype specification.

The steady states shown in Table 3 (FHF aCM, FHF vCM, SHF aCM, SHF vCM) are consistent with biological evidence, suggesting that both atrial and ventricular CMs can originate from either heart field [4,5]. Cells of the first heart field

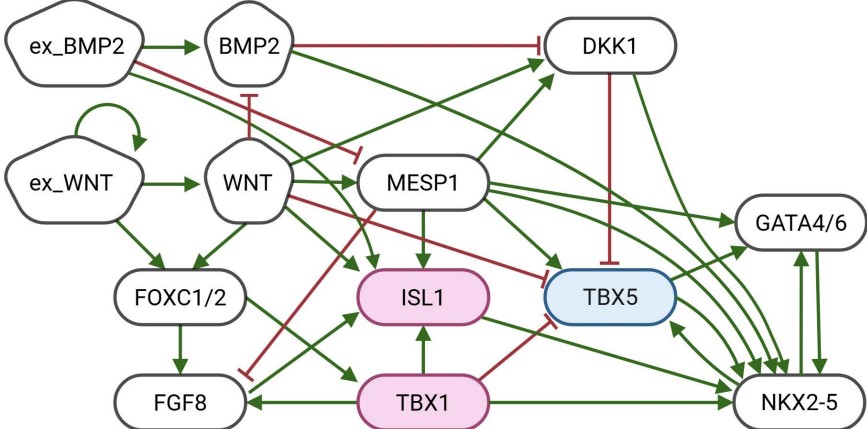

**Fig 6. Gene regulatory network underlying heart field specification, adapted from [4].** The network models differentiation of cardiac precursor cells into FHF (with central regulatory node TBX5 shown in blue) and SHF (with central regulatory nodes ISL1 and TBX1 shown in pink) fates guided by BMP and WNT signaling. Nodes are regulatory components including signaling molecules and transcription factors, with activating interactions shown as green arrows, and inhibitory interactions as red T-bars. The BN corresponding to this network is shown in S2A Table. The steady states of the heart field specification GRN are provided in Table 2.

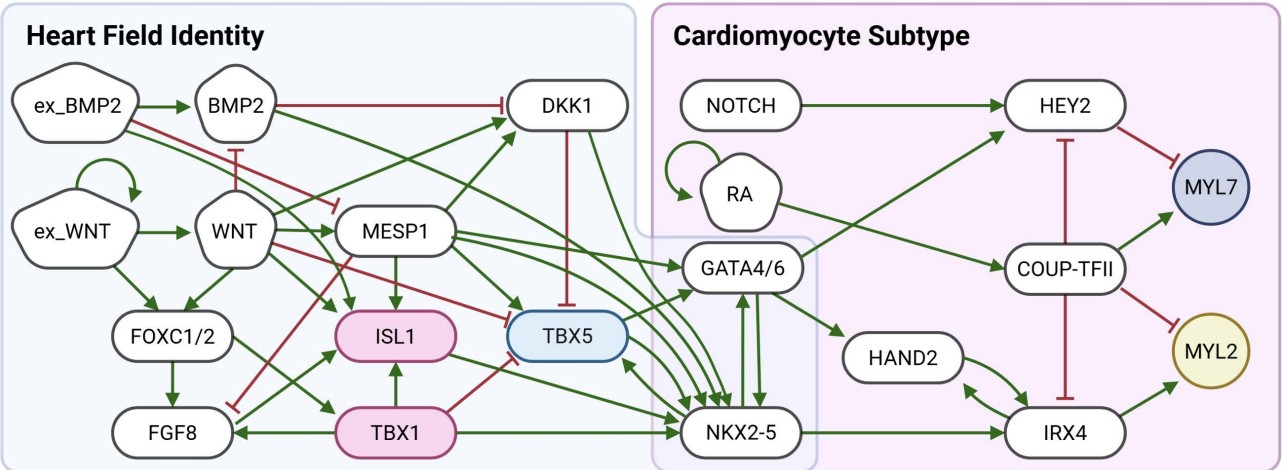

**Fig 7. The unified Gene regulatory network for heart field identity and cardiomyocyte subtype.** The nodes in the blue shaded area (left part of Fig 7) constitute the heart field GRN adapted from [4], while the nodes in the magenta shaded area are the nodes of the Cardiomyocyte GRN. The nodes in the overlapping purple area (GATA4/6 and NKX2-5) connect the models with each other. In this model, expression of TBX5 promotes FHF identity; ISL1 and TBX1 promote SHF identity, MYL7 expression indicates aCM differentiation and MYL2 expression indicates vCM differentiation. The steady states of the unified GRN are provided in Table 3.

primarily populate the left ventricle, while cells of the second heart field populate the right ventricle with both fields contributing to the atria [5]. In the model, the steady state representing FHF-derived ventricular CMs (column FHF vCM) exhibits TBX5 activity, consistent with left ventricular identity [12,22]. Conversely, SHF-derived ventricular CMs (column SHF vCM) exhibit TBX1 expression, corresponding to right ventricular identity [12,22].

Additionally, the model highlights the differential activity of key signaling pathways (WNT, BMP, RA, and NOTCH) across the different steady states. In vivo and in vitro, correct temporal activation of these pathways is crucial for guiding

**Table 2. Steady states of the heart field specification Boolean network shown in Fig 6.**

|  | Signaling molecules | | | | Transcription factors | | | | | | |
|---|---|---|---|---|---|---|---|---|---|---|---|
|  | ex_WNT | ex_BMP2 | BMP2 | WNT | FOXC1/2 | ISL1 | MESP1 | NKX2-5 | TBX1 | TBX5 | GATAs |
| FHF | 0 | 1 | 1 | 0 | 0 | 0 | 0 | 1 | 0 | 1 | 1 |
| SHF | 1 | 1 | 0 | 1 | 1 | 1 | 0 | 1 | 1 | 0 | 1 |
| null | 0 | 1 | 1 | 0 | 0 | 0 | 0 | 0 | 0 | 0 | 0 |

Each row corresponds to a steady state derived from asynchronous simulation. Columns represent node activities, where a value of 1 (green) indicates an active node, and a value of 0 (white) indicates an inactive node. During simulations, the node exogen BMP2 (ex BMP2) is always perturbed to 1 (active) as the differentiation of CMs begins at lateral plate mesoderm where ex BMP2 is always active [28]. The two steady states of the BN correspond to the two different heart fields, which are characterized by the expression of FHF marker genes (TBX5) or SHF marker genes (FOXC1/FOXC2, ISL1, TBX1) [4].

**Table 3. Steady states of the unified BN.**

| Cell types | | FHF aCM | FHF vCM | SHF aCM | SHF vCM | Null |
|---|---|---|---|---|---|---|
| Cell fate indicators | | TBX1+ MYL7+ | TBX1+ MYL2+ | TBX5+ MYL7+ | TBX5+ MYL2+ | GATA4/6- |
| Input Heart field | BMP | 1 | 1 | 0 | 0 | 1 |
| | WNT | 0 | 0 | 1 | 1 | 0 |
| Heart field Transcription factors | TBX5 | 0 | 0 | 1 | 1 | 0 |
| | ISL1 | 0 | 0 | 1 | 1 | 0 |
| | TBX1 | 1 | 1 | 0 | 0 | 0 |
| | GATA4/6 | 1 | 1 | 1 | 1 | 0 |
| Input CM Cardiomyocyte subtype Transcription factors | RA | 1 | 0 | 1 | 0 | |
| | COUP-TFII | 1 | 0 | 1 | 0 | |
| | MYL7 | 1 | 0 | 1 | 0 | |
| | IRX4 | 0 | 1 | 0 | 1 | |
| | MYL2 | 0 | 1 | 0 | 1 | |

Each column represents a steady state, corresponding to a specific cardiac differentiation outcome, and rows represent the activity of a biological factor. An entry of 1 denotes that the factor is active in the respective steady state, while 0 denotes inactivity. The steady states in columns three to six correspond to biologically relevant CM subtypes: atrial cardiomyocytes (aCM) and ventricular cardiomyocytes (vCM) derived from either the first (FHF) or second heart field (SHF). The remaining "null" steady state results from the inactivation of GATA4/6, a prerequisite for cardiac differentiation [26], therefore causing no downstream cardiomyocyte marker upregulation. Cells remaining in the "null" state are therefore speculated to be mesodermal progenitors or differentiated to another mesodermal lineage not accounted for in the model structure.

cardiomyocyte differentiation. For instance, the simulation results suggest that SHF-derived right ventricular CMs can be generated from precursor CMs, by initial activation of WNT while inhibiting BMP, followed by NOTCH activation and RA inhibition during maturation. These findings are consistent with experimental observations [8].

The presence of the "null" steady state in which GATA4/6 is inactive reflects its essential role in cardiomyocyte specification [26]. The model suggests that there is a state possible where GATA4/6 is not activated, and no cardiac markers get upregulated. These cells with a "null" state identity are therefore speculated to either be non-differentiated precursor mesoderm cells, or cells that may adopt an alternative non-cardiac cell fate not accounted for in the current network structure.

## Probabilistic simulations to predict differentiation efficiency

Probabilistic simulations of the unified model were conducted to evaluate differentiation efficiency in a heterogeneous cell population of cardiac progenitor cells. The unified BN has 21 variables, and each simulation step involved the

asynchronous selection of one variable for updating, resulting in transition probabilities of 1/21 across the corresponding Markov chain. A total of 400,000 independent simulations were performed, with 100,000 runs for each of the 4 different activity combinations of the input variables WNT and RA (as shown in the first column of Table 4). The initial values of all the other nodes were assigned with a random binary value in {0, 1}.

The results of the conducted simulations are shown in Table 4. A total of *363,695* out of *400,000* simulated cells (*90.92%*) follow the expected differentiation lineage based on the combination of the input variables (WNT, RA) that each cell is treated with. When WNT signaling was inactive (WNT = 0; 1st and 3rd row of Table 4), the network primarily produced FHF cells, as well as cells reaching the null state, suggesting a non-cardiac lateral plate mesoderm derived cell type like endothelial or smooth muscle cells [28]. Among the FHF cells, the cells exposed to RA (RA = 1) gave rise FHF aCMs. In contrast, when WNT was active (WNT = 1; 2nd and 4th row of Table 4), the model yielded SHF cells which, in absence of RA signaling (RA = 0) differentiated to SHF vCMs, in line with predicted lineage trajectories.

## Qualitative validation of the unified Boolean network

Known genetic perturbations from experimental studies were simulated using the Boolean network to evaluate the model's predictive capabilities. Knockouts were adapted to the BN by fixing the value of that node to 0 (inactive). Similarly, constitutive expression (overexpression) was modeled by setting the value of the associated node to 1 (active). The effects of each perturbation were evaluated by analyzing changes in the model's steady states and checking if resulting cell types matched experimentally observed outcomes. A summary of perturbations and their simulation results is provided in Tables 5 and 6.

In the simulations, the knockout of COUP-TFII (ii) correctly prevented atrial cardiomyocyte formation; the double knockout of NKX2.5 and HAND2 (v) successfully eliminated all ventricular cardiomyocytes; and HEY2 overexpression (vii) accurately suppressed atrial gene expression. These results align with experimental findings and demonstrate the ability of the model to mimic key gene regulatory interactions.

HAND2 knockout (i) and NKX2.5 knockout (iv), which are expected to result in the loss of right and left ventricular cardiomyocytes respectively, were only partly reproduced by the model. This can be attributed to model simplifications, particularly the exclusion of HAND1, a transcription factor strongly enriched in the left ventricle [29]. HAND1 and HAND2 exhibit functional redundancy during cardiomyocyte differentiation, with HAND1 compensating for HAND2 loss in the FHF to ensure left ventricular development in vivo [30]. Including HAND1 would therefore be redundant for modeling the general mechanisms behind the differentiation, but as a result, the model can not distinguish between right- and left ventricle-specific effects. Similarly, the effects of NKX2.5 knockout, disrupting left ventricle formation through upstream regulation of HAND1 [29], were also only partially reproduced. Despite the model's inability to fully capture the effects of these perturbations, the overall impact on ventricular cardiomyocyte loss was successfully reproduced.

Table 4. **Simulations of cardiomyocyte differentiation efficiency under varying WNT and RA input combinations.**

| Inputs | FHF aCM | FHF left vCM | SHF aCM | SHF right vCM | null |
|---|---|---|---|---|---|
| WNT = 0, RA = 0 | 0 | 81,937 | 0 | 0 | 18,163 |
| WNT = 1, RA = 0 | 0 | 0 | 0 | 100,000 | 0 |
| WNT = 0, RA = 1 | 81,858 | 0 | 0 | 0 | 18,142 |
| WNT = 1, RA = 1 | 0 | 0 | 100,000 | 0 | 0 |

Each row represents a simulation condition in which 100,000 independently initialized cells were exposed to a specific combination of WNT and RA input signals. Columns indicate the number of cells that differentiated into each of the five steady states. In the first row (WNT = 0, RA = 0), 81,937 cells (over 80%) differentiated FHF derived left ventricular cardiomyocytes, while 18,163 cells transitioning into the null state due to insufficient input activation. In the second row (WNT = 1, RA = 0), all cells are differentiated into SHF derived right ventricular cardiomyocytes. The results of the 3rd and the 4th row of Table 4 can be interpreted accordingly, and demonstrate that WNT activation drives differentiation toward SHF-derived lineages, while RA signaling promotes atrial fate.

**Table 5. Overview of cardiomyocyte differentiation perturbations in literature.**

| id | Perturbed Node(s) | Literature |
|---|---|---|
| (i) | HAND2 | HAND2 knockout results in no right ventricle [31–33]. |
| (ii) | COUP-TFII | COUPTF-II knockout results in no atrial cardiomyocytes [34]. |
| (iii) | TBX5 | TBX5 knockout results in no atrial cardiomyocytes [35]. |
| (iv) | NKX2.5 | NKX2.5 knockout results in no left ventricle [29]. |
| (v) | NKX2.5, HAND2 | NKX2.5 with HAND2 knockout gives no ventricle [31]. |
| (vi) | HEY2 | HEY2 knockout mouse cannot develop a normal left ventricle. HEY2 knockout mice get ectopic atrial gene expression in the left ventricle [6,36]. |
| (vii) | HEY2 | Overexpression of HEY2 in atrial cardiomyocytes represses atrial genes [6,37,38]. |

Each row lists a published genetic perturbation (knockout or overexpression) that affects cardiomyocyte subtype formation. Column 1 provides the identifier (id) of the perturbation. Column 2 lists the perturbed node(s) which can be either a knockout or an overexpression. Column 3 includes the literature where the corresponding perturbation is referenced.

**Table 6. Simulation results for in silico perturbations compared to experimentally observed results.**

| | WT | | (i) HAND2 KO | | (ii) COUP-TFII KO | | (iii) TBX5 KO | | (iv) NKX2-5 KO | | (v) NKX2-5 KO HAND2 KO | | (vi) HEY2 KO | | (vii) HEY2 Overexpression | |
|---|---|---|---|---|---|---|---|---|---|---|---|---|---|---|---|---|
| | Exp | Sim | Exp | Sim | Exp | Sim | Exp | Sim | Exp | Sim | Exp | Sim | Exp | Sim | Exp | Sim |
| WNT=0, RA=1 (FHF aCM) | | | | | | | | x | | | | | | | | |
| WNT=0, RA=0 (FHF vCM) | | | | x | | | | | | | | | | x | | |
| WNT=1, RA=1 (SHF aCM) | | | | | | | | x | | | | | | | | |
| WNT=1, RA=0 (SHF vCM) | | | | | | | | | | x | | | | | | |
| Succes rate | 100% | | 75% | | 100% | | 50% | | 75% | | 100% | | 75% | | 100% | |

Cardiomyocyte subtypes appear under specific signaling combinations (column 1). Literature reports that under genetic perturbations, some cardiomyocyte subtypes do not appear anymore (as seen in Table 5). The table compares expected differentiation outcomes from literature (Exp) with simulated steady states (Sim) for seven genetic perturbations. Light green in the expected column (Exp) indicates that a cell type is reported to appear in the literature with that signaling combination (WNT, RA), while white denotes its absence. In the simulated column, dark green indicates that a steady state for a cell type is reached during simulations with that signaling combination (WNT, RA), whereas gray indicates that the steady state is not reached. If both the expected and simulated column agree that a cell type does or does not appear under the signaling combination, the simulation is considered a success. Mismatches between expected and simulated outcomes are unsuccessful and marked with a cross. The model reproduced 27 of 32 combinations of signaling and perturbations (~84%).

Together, these results demonstrate that the unified Boolean network captures the regulatory logic of heart-field-specific cardiomyocyte differentiation. In addition to reproducing perturbation outcomes, the model also highlights how signaling combinations determine the cardiac lineage trajectories. In summary, the simulations suggest that cardiomyocyte cell fate is orchestrated by a small but highly influential set of factors, primarily WNT and BMP during heart field specification

followed by RA during the cardiomyocyte subtype commitment, consistent with in vivo and in vitro observations. These findings show the robust logic behind the cardiac developmental system.

## Discussion

This paper presents a Boolean GRN that integrates heart field specification with the differentiation of committed cardiomyocytes (CMs) to atrial and ventricular subtypes. The unified model incorporates key signaling pathways and transcription factors implicated in both cardiomyocyte differentiation stages. Attractor analysis identifies four steady states corresponding to heart-field-specific cardiomyocyte identities: FHF-derived atrial CMs, FHF-derived ventricular CMs, SHF-derived atrial CMs, and SHF-derived ventricular CMs. This work expands upon earlier models focused exclusively on heart field formation [4] by simulating downstream CM subtype commitment.

### Regulatory robustness and switch-like behavior revealed through Boolean network extension

The attractors generated by the model reproduce known gene expression profiles for atrial and ventricular fates, including MYL7 and COUPTFII for atrial identity and MYL2 and HEY2 for ventricular identity. Retinoic acid (RA) signaling acts as the key regulatory input driving the switch between these identities, consistent with in vivo and in vitro differentiation studies [13,23].

To investigate the concentration dynamics of this regulatory switch, the cardiomyocyte subtype Boolean network was converted into a Boolean network extension. In this framework, node activity takes values across the continuous interval [0, 1], and update functions are reformulated as real-valued expressions. Analysis of the Boolean network Extension revealed that the decision between atrial and ventricular specification depends on the RA activity crossing a threshold. This supports the idea that varying RA concentration can instruct distinct cardiac subtype fates, consistent with its proposed morphogenic function in cardiomyocyte differentiation [23]. There are several co-factors and enzymes that influence RA signaling and RA-dependent gene expression, including COUPTFII [23]. Therefore, the RA concentration to cross the signaling threshold in vitro should be calibrated based on the experimental context.

### Model validation using knockout and overexpression perturbations

To validate the predictions of the network, the model was tested against seven perturbation experiments described in the literature. Three out of seven perturbations were fully reproduced, such as the suppression of atrial fate upon COUP-TFII knockout [34] and repression of atrial gene expression under HEY2 overexpression [6,37,38]. Two perturbations were partly reproduced, and two perturbations were not captured. These results support the model's capacity to capture core regulatory dependencies driving CM subtype identity.

The discrepancies in some of the model predictions reflect either missing regulatory links or simplifications inherent in a Boolean logic framework. For instance, in the case of TBX5 knockout [35], the model predicted no change in steady states except the deactivation of TBX5 itself. This result is consistent with experimental observations showing that MYL7 expression persists in TBX5-deficient mice, but it fails to capture the reduction in atrial cardiomyocyte output mediated by loss of TBX5-regulated genes such as ANF and CX40 [35]. In the context of this knockout, MYL7 can therefore not represent atrial genes as it does for the wild type.

The HEY2 knockout simulation also deviated from biological expectations. HEY2 loss has been shown to induce ectopic MYL7 expression in ventricular cells [6], but this behavior was not reproduced. The discrepancy arises from the Boolean logic constraints, which prevent a steady state in which MYL2 and MYL7 are both active. In contrast, the BNE analysis, which allowed continuous activation values, did permit intermediate MYL7 activity in ventricular states, and may therefore have represented the effects of this knockout more accurately.

Other mismatches likely result from the simplified network topology. In these cases, the mechanisms underlying the effects of a perturbation are not captured because the relevant regulatory factors are not included in the model.

Incorporating such factors in future versions could improve the model's predictive powers, especially under more complex or combined perturbations. However, expanding the network could cause redundancy of nodes and feedback loops that enhance the robustness of the system. This would make it harder to learn from the model, as simulations may be less responsive to perturbations like knockdown or overexpression. In effect, expanding the model may paradoxically lead to a reduction of predictive power and model sensitivity [16].

## Justification for Boolean modelling approach

Differential equation-based models can describe time-dependent processes and concentration-dependent interactions, but they require detailed kinetic parameters and quantitative data. Such information is rarely available for large gene regulatory networks involved in embryonic development [4,16]. Boolean models provide a simplified yet effective alternative for modeling gene regulatory networks when mainly qualitative information is available [15,16]. In this study, a Boolean framework was applied to simulate the differentiation of heart-field-specific cardiomyocyte subtypes. All regulatory interactions in the model were derived from published experimental studies, without requiring parameter estimation or assumptions about reactions kinetics. This approach enables qualitative predictions of regulatory behavior using a binary representation of activity.

The Boolean logic approach did however have limitations for modelling heart-field-specific cardiomyocyte differentiation. Two networks that biologically are consecutive were united and steady states were computed asynchronously. Because of this, nodes determining cardiomyocyte identity are already interacting while the heart field decision is still being made. As a result, cardiomyocyte regulators that are heart field specific (e.g., HAND1) cannot be included in the model. Using Boolean logic also caused effects of some knockouts to not be captured correctly. Some knockouts biologically cause ectopic expression of marker genes in other tissues (e.g., MYL7 expression in MYL2-expressing ventricular cardiomyocytes after HEY2 knockout). The Boolean logic does not allow for a steady state to exist where two separate markers are both expressed in the same cell, and these knockouts will therefore not be successfully represented. Although the Boolean modelling approach allows for insights in the system without detailed data on parameters, it lacks implementation of timing and concentrations which are important properties during differentiation.

Boolean networks have previously been used for understanding the regulatory mechanisms behind differentiation of multiple different cell types. This is done either by inferring GRN structure of a system from a single cell or bulk dataset (RNA-seq or qRT-PCR) with an algorithm (e.g., [39–41]), or by manually assigning Boolean rules to literature-reported observations (e.g., [4,15,42,43]). The approach in this manuscript is similar to the second category, but extends it in two ways. Instead of modelling a single lineage decision, we couple two consecutive differentiation stages in one Boolean Network, to see how heart-field-specific cardiomyocytes can be generated. We also couple our GRN to external signaling inputs to gain additional insights on how the GRN activities can be influenced to steer differentiation towards a specific cell type. Other modelling modalities can also be used to predict signaling combinations with additional information on the effects of signaling strength (such as timed automata, e.g., [44]; and ODE-based modelling, e.g., [45,46]), something Boolean network predictions lack. Still, these methods require a tremendous amount of data for parameter optimization, whereas here it's shown that Boolean Networks can already provide applicable insights from a simple dynamic approach.

## Broader implications and future directions

The model presented provides an integrated Boolean Network that connects early heart field specification with differentiation towards atrial and ventricular subtypes. The dual-stage integration shows how early signals influence the subtype identities, providing a foundation as a hypothesis-generating tool for understanding the emergence of heart-field-specific cardiomyocytes. Insights from the model can aid the development of in vitro differentiation protocols aimed at producing specific CM types.

The model reproduces known expression patterns and perturbation phenotypes but could be further enhanced through additional validation steps. Quantitative comparison of the model predictions against a transcriptomic dataset requires an optimized experiment that uses the same set of signaling factors as in the model, applied to the corresponding cell types. Setting up and optimizing such an experiment might be timely and costly but would significantly strengthen conclusions from the model. New knockout experiments can be essential in extending the model to enhance its predictive accuracy. Such validations would support the refinement of the network into a better hypothesis-generating tool.

Ultimately, the model offers a promising foundation for understanding the control of cardiac development. The ability to reproduce key regulatory behaviors with minimal assumptions is especially valuable in a data-limited context. With integration of new datasets and modelling formalisms, the model can be progressively refined to capture more dynamic, spatial and mechanical aspects of heart development, to bring it closer to applications in disease modeling and cell therapy.

## Supporting information

**S1 Table. Extended evidence of the connections constituting the cardiomyocyte subtype GRN.** The listed sources form the basis of the interactions that make up Fig 4.
(DOCX)

**S2 Table. Boolean network description of the models.** The Table includes the description of the S1A) Heart Field specification S1B) Cardiomyocyte specification and S1C) Unified differentiation network.
(DOCX)

## Acknowledgments

This work is part of work conducted within the Heart Engine consortium involving the groups of Prof. Dr. J. Leijten, Prof Dr. R. Passier, and Dr. Ing. J. N. Post (University of Twente, The Netherlands), and Prof. Dr. G.J.C. Veenstra (Radboud University, The Netherlands). The study has benefited from discussions with the following: Stefano Schivo from the Open Universiteit, Heerlen, the Netherlands, Lucas F. Jansen Klomp, Xinqi Yan from the University of Twente, the Netherlands, and Rebecca R. Snabel, Gert Jan C. Veenstra from Radboud University, the Netherlands. This manuscript is a revised and expanded version of the preprint "Molecular mechanisms of heart field specific cardiomyocyte differentiation- a computational modeling approach" previously uploaded on bioRxiv (https://doi.org/10.1101/2024.12.19.629328). The description of the materials, methods, and core results is based on that original draft.

## Author contributions

**Conceptualization:** Ricco Zeegelaar, Georgios Argyris, Janine N. Post.

**Data curation:** Ricco Zeegelaar, Georgios Argyris.

**Formal analysis:** Ricco Zeegelaar, Georgios Argyris.

**Funding acquisition:** Janine N. Post.

**Investigation:** Ricco Zeegelaar, Georgios Argyris.

**Methodology:** Ricco Zeegelaar, Georgios Argyris.

**Project administration:** Georgios Argyris, Janine N. Post.

**Resources:** Ricco Zeegelaar, Georgios Argyris, Janine N. Post.

**Software:** Ricco Zeegelaar, Georgios Argyris, Janine N. Post.

**Supervision:** Janine N. Post.

**Validation:** Ricco Zeegelaar, Georgios Argyris.

**Visualization:** Ricco Zeegelaar, Georgios Argyris.

**Writing – original draft:** Ricco Zeegelaar, Georgios Argyris.

**Writing – review & editing:** Ricco Zeegelaar, Georgios Argyris, Janine N. Post.

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
