## [Decision Letter · Decision Letter 0]

22 Sep 2025

Dear Dr. Zeegelaar,

Thank you for submitting your manuscript to PLOS ONE. After careful consideration, we feel that it has merit but does not fully meet PLOS ONE’s publication criteria as it currently stands. Therefore, we invite you to submit a revised version of the manuscript that addresses the points raised during the review process.

Specifically, the reviewers have significant concerns about weaknesses in biological validation, model justification, and clarity of result presentation

We look forward to receiving your revised manuscript.

Kind regards,

Federica Limana

Academic Editor

PLOS ONE

Journal Requirements:

2. Please ensure that you refer to Figure 1 in your text as, if accepted, production will need this reference to link the reader to the figure.

3. Please include a copy of Table 6 which you refer to in your text on page 11.

Reviewers' comments:

Reviewer's Responses to Questions

**Comments to the Author**

1. Is the manuscript technically sound, and do the data support the conclusions?

Reviewer #1: Yes

Reviewer #2: Partly

2. Has the statistical analysis been performed appropriately and rigorously?

Reviewer #1: Yes

Reviewer #2: N/A

3. Have the authors made all data underlying the findings in their manuscript fully available?

Reviewer #1: Yes

Reviewer #2: Yes

4. Is the manuscript presented in an intelligible fashion and written in standard English?

Reviewer #1: Yes

Reviewer #2: Yes

Reviewer #1: The manuscript presents a computational Boolean network (BN) model that integrates early heart field specification with downstream differentiation of cardiomyocytes (CMs) into atrial and ventricular subtypes. This dual-stage modeling approach is innovative and highly relevant to regenerative medicine, as it attempts to bridge the gap between developmental biology and therapeutic applications. The work is ambitious, clearly structured, and supported by extensive literature. However, while the study is promising, several issues limit its impact in its current form. The manuscript requires major revisions to address weaknesses in biological validation, model justification, and clarity of results presentation.

1. The exclusion of key regulators (e.g., HAND1, PITX2, ANF, CX40) undermines the model’s ability to reproduce knockout phenotypes. Please justify these exclusions or consider expanding the network to include them.

2. Reliance on MYL7 as the sole atrial marker is insufficient. Incorporating multiple atrial markers would improve biological accuracy.

3. The model only partially reproduces published knockout/overexpression studies. Quantitative validation against transcriptomic datasets (e.g., single-cell RNA-seq of differentiating cardiomyocytes) would significantly strengthen the conclusions.

4. Tables summarizing perturbations should include a quantitative measure of agreement (e.g., percentage of expected phenotypes reproduced).

5. Many findings (e.g., RA as a switch, roles of WNT/BMP) are already well established. The novelty lies in integrating early and late differentiation stages, and this distinction should be clearly emphasized.

6. The interpretation of the “null” state as alternative fates is speculative and requires either supporting evidence or a more cautious presentation.

7. Figures 6–7 (network schematics) are dense and difficult to interpret. Simplify or reorganize them, perhaps highlighting only key regulators while relegating secondary interactions to Supplementary Figures.

8. The discussion should better frame the model as a hypothesis-generating tool, not yet predictive enough for guiding clinical differentiation protocols.

9. Limitations of the Boolean logic approach (e.g., inability to capture simultaneous MYL2 and MYL7 activation in HEY2 knockout) should be more explicitly acknowledged.

10. Provide future directions: validation with transcriptomics, incorporation of temporal/spatial aspects, or hybrid Boolean–ODE modeling.

11. Ensure consistency in terminology (“heart field” instead of “heat field” in Table 2 caption).

12. References: ensure consistent formatting and check for missing details in some citations.

13. Improve figure captions: they should interpret results biologically, not just describe content.

Reviewer #2: This paper addresses an important problem of predicting heart field of the cardiomyocytes when they are differentiated in vitro. This problem is of general interest to both system biologists in terms of cell-fate modeling and experimentalist working with iPSC-derived cardiomyocytes. However, the manuscript has some major issues detailed below:

(please note that there were no pages, section, or line numbers, so the page with the abstract was assumed to be 1)

1. Right from the introduction, it is not fully clear if the validation of this model has been given proper weight. It would be useful to provide an introduction to how this type of model framework can be validated. (last paragraph page 3)

2. The idea behind the model dynamics for this paper is rather interesting, but it would be really nice to provide a bit of background here for the more general audience of PLOSOne. As it is, the paper is not very accessible outside the field of synthetic-biology. (page 4 bottom).

3. Page 5 - knockout validation paragraph. This could use some more detail as it wasn't clear from this description how the simulation results would be compared back to the experimental data for proper validation.

4. The equation derivation on page 8, seemed out of context and a bit confusing. It is not clear to a general reader how these equations provide insights into the dynamics. Is something known about this dynamics experimentally? Can the authors provide plots to help visualize what they mean? Figure 5 does not have enough context to understand how this prediction is helpful without something similar from experiments or a more comprehensive explanation.

5. The table descriptions some times get confusing as they refer to column numbers in tables with split columns (page 10).

6. The abbreviations in the tables should be more consistently defined (example aCM is not defined in the table on page 10).

7. Table 3 has one of the most interesting results of the paper, showing an ability to predict cell populations. It should be possible to compare these results to results from single cell RNA-seq experiments, but there is no mention of validating the model in this way. This would be an actual quantitative validation, which would be very valuable, and should be discussed in length. (page 11)

8. The validation on page 12 is purely qualitative, and the logic for why it is so instead of a quantitative validation is confusing. It's confusing that "light green indicates that a cell type is predicted to appear" for experimental work. Is it dominant? What does it mean predicted in such a context?

9. Page 13 - it would be nice if the discussion section presented this work in the context of other works that predicted cell fate during differentiation.

10. The first paragraph of model validation on page 13 is overstated - only 3 of the 7 fully agreed (and qualitatively at that).

Minor

- There are two Table 2s and a missing Table 6. It was rather hard to know what referenced what as a result

- Page 10 last line, the periods were probably meant to be commas in the large numbers of simulations.

**Do you want your identity to be public for this peer review?** For information about this choice, including consent withdrawal, please see our Privacy Policy

Reviewer #1: No

Reviewer #2: No

---

## [Author Response · Author response to Decision Letter 1]

17 Nov 2025

We have responded to the reviewer and editor comments in the file by the name of "Response to Reviewers". We thank the reviewers and editor again for their insights and adjusted the manuscript accordingly.

---

## [Decision Letter · Decision Letter 1]

17 Dec 2025

Molecular insights into heart field-specific cardiomyocyte differentiation - a computational study

PONE-D-25-32229R1

Dear Dr. Zeegelaar,

We’re pleased to inform you that your manuscript has been judged scientifically suitable for publication and will be formally accepted for publication once it meets all outstanding technical requirements.

Kind regards,

Federica Limana

Academic Editor

PLOS One

Additional Editor Comments (optional):

Reviewers' comments:

Reviewer's Responses to Questions

**Comments to the Author**

Reviewer #1: All comments have been addressed

Reviewer #2: All comments have been addressed

2. Is the manuscript technically sound, and do the data support the conclusions?

Reviewer #1: (No Response)

Reviewer #2: Yes

3. Has the statistical analysis been performed appropriately and rigorously?

Reviewer #1: (No Response)

Reviewer #2: Yes

4. Have the authors made all data underlying the findings in their manuscript fully available?

Reviewer #1: (No Response)

Reviewer #2: Yes

5. Is the manuscript presented in an intelligible fashion and written in standard English?

Reviewer #1: (No Response)

Reviewer #2: Yes

Reviewer #1: The authors have addressed my concerns. The revised manuscript can now be accepted in its present form.

Reviewer #2: This paper will be of interest to PLOS One readers. From the previous review, all my comments have been addressed.

**Do you want your identity to be public for this peer review?** For information about this choice, including consent withdrawal, please see our Privacy Policy

Reviewer #1: No

Reviewer #2: No

---

## [Editor Report · Acceptance letter]

PONE-D-25-32229R1

PLOS One

Dear Dr. Post,

I'm pleased to inform you that your manuscript has been deemed suitable for publication in PLOS One. Congratulations! Your manuscript is now being handed over to our production team.

Kind regards,

on behalf of

Dr. Federica Limana

Academic Editor

PLOS One